# Nutrition in Pediatric Intensive Care: A Narrative Review

**DOI:** 10.3390/children9071031

**Published:** 2022-07-11

**Authors:** Milan Kratochvíl, Jozef Klučka, Eva Klabusayová, Tereza Musilová, Václav Vafek, Tamara Skříšovská, Jana Djakow, Pavla Havránková, Denisa Osinová, Petr Štourač

**Affiliations:** 1Department of Paediatric Anaesthesiology and Intensive Care Medicine, University Hospital Brno and Faculty of Medicine, Masaryk University, Kamenice 5, 625 00 Brno, Czech Republic; milo.kratochvil@gmail.com (M.K.); jozoklucka@gmail.com (J.K.); eva.klabusayova@gmail.com (E.K.); musilovate@gmail.com (T.M.); vaclav.vafek@hotmail.com (V.V.); tamara.skrisovska@gmail.com (T.S.); pa.havrankova@gmail.com (P.H.); petr.stourac@gmail.com (P.Š.); 2Department of Simulation Medicine, Faculty of Medicine, Masaryk University, Kamenice 5, 625 00 Brno, Czech Republic; 3Paediatric Intensive Care Unit, NH Hospital Inc., 268 01 Horovice, Czech Republic; 4Department of Anaesthesiology and Intensive Care Medicine, The Donaustadt Clinic Republic, Lango Bardenstraße 122, 1220 Vienna, Austria; 5Department of Anaesthesiology and Intensive Care Medicine, University Hospital Martin and Jessenius Medical Faculty in Martin, Comenius University Bratislava, Malá Hora 4a, 03601 Martin, Slovakia; denisa.osinova@gmail.com

**Keywords:** nutrition, nutrition support, enteral feeding, parenteral nutrition, pediatric, child, intensive care

## Abstract

Nutrition support in pediatric intensive care is an integral part of a complex approach to treating critically ill children. Smaller energy reserves with higher metabolic demands (a higher basal metabolism rate) compared to adults makes children more vulnerable to starvation. The nutrition supportive therapy should be initiated immediately after intensive care admission and initial vital sign stabilization. In absence of contraindications (unresolving/decompensated shock, gut ischemia, critical gut stenosis, etc.), the preferred type of enteral nutrition is oral or via a gastric tube. In the acute phase of critical illness, due to gluconeogenesis and muscle breakdown with proteolysis, the need for high protein delivery should be emphasized. After patient condition stabilization, the acute phase with predominant catabolism converts to the anabolic phase and intensive rehabilitation, where high energy demands are the keystone of a positive outcome.

## 1. Introduction

The initial protective body reaction to critical illness is based on simple phylogenetic targets aimed at oxygen and energy (E) delivery to the vital organs. During the acute phase (e.g., shock), multiple endocrine, immunology and metabolic reactions are initiated to minimize the initial insult. This stage is usually associated with compromised energy and protein delivery; moreover, energy utilization failure can also appear. The maximum effort for adequate E delivery leads to glycogenolysis and gluconeogenesis (with muscle breakdown), where all substrates (including amino acids) are being used. These metabolic disturbances in the acute phase of critical illness are associated with insulin resistance, stress hyperglycemia and are responsible for the shift to catabolism [1,2]. Although initially lifesaving, these metabolic disturbances can lead to profound catabolism, severe malnutrition and an increase in morbidity and mortality if prolonged over the acute phase of illness [1,2,3]. Initial nutritional assessment and early nutritional support (after initial stabilization, within 48 h from admission) is therefore an integral part of pediatric critical care [1,4]. Nutritional support should be individualized on a case-by-case basis, due to the possible high malnutrition rate at pediatric intensive care unit (PICU) admission (up to 25%) [1,5,6,7] with close monitoring of target E and protein delivery over the PICU stay. According to the pathophysiology of critical illness during the initial state with muscle breakdown and possible protein wasting, nutritional support aims to minimize the negative protein balance. The excessive E delivery (overfeeding) can be harmful in the acute phase due to ineffective substrate utilization. Protocol-based nutritional support in PICU is recommended, and several guidelines have been developed over the past decades, with the European Society of Pediatric and Neonatal Intensive Care (ESPNIC) [1], European Society for Clinical Nutrition and Metabolism (ESPEN) [4,8] and American Society for Clinical Nutrition and Metabolism (ASPEN) and Society of Critical Care Medicine (SCCM) [9] being the most adopted and cited. The impact of proper nutritional support is highlighted by its association with the infection rate, length of stay and mortality [10,11,12,13]. When considering the nutritional support for a pediatric critically ill patient, the main principles remain the same as in adults (early nutritional status assessment, preference of oral intake, preference of enteral feeding over parenteral feeding, etc.). However, several dissimilarities should be esteemed, such as different nutritional status assessment and different E targets and nutrient composition for various age categories. The aim of this narrative review was to describe the step-by-step approach for nutritional support of the general population of critically ill pediatric patient in the PICU. This review does not focus on the more specific population of very small infants and neonates.

## 2. Nutritional Status Assessment

As previously described, a high prevalence of malnutrition can be diagnosed upon PICU admission [14,15,16,17]. Due to the strong association of malnutrition (including obesity) with negative clinical outcomes, the early detailed assessment of nutritional status is recommended as a part of PICU admission [18]. Nutritional risk in critically ill (NUTRIC) patients [4,19] and nutritional risk screening (NRS 2002) [4,20] have been validated and adopted for adult patients. For pediatric patients, weight and height/length (in children < 2 years of age) and z score for body mass index (BMI) for age, mid upper-arm circumference or weight for age can be used [1,18]. For children under 36 months, head circumference should also be monitored [18]. The anthropometric measurements should be performed repeatedly (according to the nutrition protocol) during the whole PICU stay to indirectly evaluate the effect of nutritional interventions. Besides a close physical examination, complete dietary history and functional status prior to PICU admission should be assessed [18]. Nutritional assessment generally requires special training, clinical expertise and more time to complete. Such a formal nutritional assessment typically requires a dietitian, nutrition specialist or other qualified health care provider. On the other hand, a quick way to assess nutritional status is nutrition screening tools, which are expected to be able to be administered quickly (i.e., less than 15 min) and do not require special training for the caregiver. The complete assessment can then be finished later, especially in high-risk patients. Several screening tools are available in the literature; however, none of them have been validated in the PICU [21,22,23,24]. The Screening Tool for Risk of Impaired Nutritional Status and Growth (STRONGKids) appears to be a suitable admission screening test for malnutrition in PICU [25]; however, it is unable to evaluate children with abnormal anthropometrics [18]. A recent multicenter study of inpatients in 14 hospitals across 12 European countries found that three of these more commonly used screening tools showed a fair amount of variability in the classification of malnutrition risk, and each tool failed to identify a proportion of children with subnormal anthropometric measures [25]. Therefore, it seems reasonable to combine screening tools with anthropometric measurements and reassess them regularly during the PICU stay. Other possible screening methods for malnutrition are currently used mainly for research purposes and have not been validated, such as ultrasound, computed tomography (CT) and bioelectrical impedance [4,26,27,28]. However, ultrasound muscle mass and density screening seem to be promising methods for the future due to their wide accessibility, minimal financial requirements and minimal staff’s experience demands. 

## 3. Nutritional Support Initiation

Nutritional (enteral) support should be initiated as soon as possible. Unless contraindicated, enteral feeding should be initiated during the first 24 h after admission [1]. The preferred route of delivery is gastric, ideally oral. In the case of an inability to be fed naturally (swallowing disorders, risk of aspiration), a gastric tube should be used. In small infants (e.g., under 6 months of age), breast feeding, or tube feeding with breast milk, should be encouraged under any circumstances. In very small critically ill infants, breast milk can be used as the exclusive source of nutrients. The classically cited contraindications for enteral feeding are tight bowel stenosis, severe diarrhea, severe malabsorption, gut ischemia, decompensated shock, severe acidosis and/or hypoxemia, uncontrolled upper gastrointestinal (GI) bleeding, bowel obstruction, abdominal compartment syndrome and high-output fistula [4,29]. 

## 4. Optimal Energy and Protein Delivery for a Critically Ill Child

The optimal energy delivery varies significantly over the PICU stay (lower E demands in the acute state, higher in the recovery state). A simple formula (such as 20–25 kcal/kg in the acute stage and 30–35 kcal/kg in recovery in adults) [4,9,30,31] will not fit all age categories, as E demands per kilogram are higher in neonates and infants. The ideal method for the individualized E demand estimation is considered the indirect calorimetry (IC) [4,18]. The availability of IC in PICUs is, however, still limited. In the case of IC’s absence, the Schofield, Food Agriculture Organization, World Health Organization (WHO) or United Nations University equations without additional risk factors can be used for optimal energy delivery calculation (see Table 1). [18], where the Schofield equation (age, gender, height and weight) is preferred by the ESPNIC [1]. Another recommended method for optimal/individualized E-target estimation is the volumetric measurement of VO_2_ and/or VCO_2_ [4,12,32]. The Harris–Benedict equation, which has been used for several decades, is not recommended in adult or pediatric intensive care anymore. Although not well-defined for the pediatric population, in the acute phase of adult critical illness, a hypocaloric target of 70% energy expenditure should be used [4].

The aim of protein delivery is to avoid a negative protein balance (high risk in the early phase). For pediatric patients, the protein delivery should be minimally 1.5 g/kg/day (even higher in patients with respiratory failure and burns) [1,18]. However, data confirming the positive influence of higher protein delivery (over 1.5 mg/kg/day) in the acute phase of critical illness on overall clinical outcome are currently missing [1,18]. The protein (or amino acid) delivery should not exceed 3 g/kg/day or 2 g/kg/day in stable adolescents [33]. Despite insufficient data for pediatric patients, in adult patients, hypocaloric nutrition (not exceeding 70% of energy target) during the acute phase of critical illness with a progressive increase over 3 days up to 100% of the E target in the case of indirect calorimetry is used. On the contrary, if the energy need is estimated using predictive equations, hypocaloric nutrition is recommended (not exceeding 70% of calculated energy need) during the first week of critical illness [4]. In the acute phase, the protein E to nonprotein E ratio (defined by grams of nitrogen to nonprotein kcal ratio) should be higher (e.g., 1 gN:80 nonprotein kcal) compared to the late phase, where anabolic process prevails (ration 1 gN:130 or higher).

The E and protein delivery aims for 66% (up to 100%) of calculated targets in the first seven days after PICU admission [18], preferentially by enteral route, if possible. Parenteral nutrition or supplemental parenteral nutrition can be initiated in patients with severe malnutrition or individually in patients not at all tolerating enteral nutrition during the first week of their PICU stay [18]. However, especially in previously healthy, well-fed children, parenteral nutrition can be safely delayed until the eighth day [34]. Based on the results of the multicentric observational trial PEPaNIC, where late parenteral nutrition was associated with improved clinical outcomes and reduced incidence of associated complications in the pediatric PICU patients [34], the ESPEN guidelines recommend postponing parenteral nutrition independent of nutritional status in the first week of PICU stay [1,5,18,35].

## 5. Macronutrients

### 5.1. Proteins

Low protein delivery in combination with proteolysis and a negative protein balance remains a common problem in PICU settings. The minimal protein dose of 1.5 mg/kg/day is recommended by several guidelines [1,4,18]; however, a much higher dose may be needed to maintain a neutral or even positive protein balance. The protein/amino acid dose should, however, be limited to 3 g/kg/d in neonates and infants and 2 g/kg/d in stable adolescents [33]. Proteins are polymers of amino acids. The amino acids can be divided into three groups: essential, semi-essential and nonessential. The essential amino acids cannot be synthesized in vivo; therefore, they must be supplemented by oral/enteral or parenteral form. Semi-essential (conditionally essential) amino acids can become essential in several clinical settings: infancy, critical illness, trauma, burns and others. The amino acid classification is listed in Table 2. The so-called pharmaconutrition (e.g., glutamine, arginine supplementation) is not currently recommended for PICU settings [1]. However, in preterm neonates, arginine (necrotizing enterocolitis prevention), tyrosine and cysteine supplementation can be associated with improved outcomes [33]. The energy content of 1 g of protein is around 4 kcal (17 kJ) and is equal to 1 g of carbohydrates. E delivery in the form of proteins/amino acids in the mixed enteral or parenteral formula should be between 15 and 30%.

### 5.2. Carbohydrates

Carbohydrates contain 4 kcal/1 g and should represent between 40 and 60% of E intake. In enteral feeding, the carbohydrate can be delivered as monosaccharides (glucose, fructose), oligosaccharides (lactose, mannose, dextrins) and polysaccharides (starch). Rapid glycolysis after carbohydrate delivery leads to a breakdown of oligomers and polymers to monomers, where glucose is the main final carbohydrate E substrate transported to the tissues. In the case of parenteral nutrition, the carbohydrates are delivered in the form of 5%, 10%, 20%, 40% or even 50% glucose (dextrose) solutions. The preferred glucose concentration is limited by the route of delivery (a peripheral intravenous line can be used up to 850 mosmol/L ≈ 12.5% glucose solution or lower) and fluid intake (higher concentration preferred in patients with limited daily fluid intake). The carbohydrate/glucose delivery should meet the metabolic/energy demands but should not lead to overfeeding, which might cause hyperglycemia and excessive CO_2_ production. Glucose/carbohydrate delivery aiming to meet the basal E needs should be between 0.5 and 10 mg/kg/min (0.7–14 g/kg/day) [36], with a lower threshold in older children and the acute phase of critical illness, higher in younger patients and later (recovery) in the course of ICU stay. The so-called tight glucose control (aiming to maintain the laboratory glycemia reference levels) is associated with worse outcomes, and it is not recommended [37]; however, hypoglycemia (<2.5 mmol/L) and/or hyperglycemia (>10 mmol/L) should be aggressively treated in all patients [36]. Blood glucose can be monitored by validated point-of-care (POCT) glucometers from peripheral blood or capillary blood samples in a bedside manner; however, in patients with shock and high vasopressor requirements, an arterial blood sample or central venous blood sample should be preferred.

### 5.3. Lipids

Lipids are an integral part of complex enteral and/or parenteral nutrition. Lipids represent a significant E substrate over 2 times denser than carbohydrates or proteins, as 1 g of lipid contains 9 kcal of E. Lipids are also integral parts of cell membranes (phospholipids) and play a role in the signaling process, hormone synthesis, vitamin metabolism (lipid-soluble vitamins A, D, E, K) and other vital biochemical processes. Similarly to amino acids, fatty acids, the fundamental part of fat tissue can be divided into essential and nonessential, based on the possibility of in vivo synthesis in the body of the patient. Essential fatty acids (EFAs) are polyunsaturated omega-3 (ω-3) (alpha-linolenic acid) and omega-6 (ω-6) acids (linoleic acid), which should be supplemented in nutrition. Several fatty acids, such as eicosatetraenoic acid (EPA) and docosahexaenoic acid (DHA), can be considered as conditionally essential due to a limited conversion capacity, and their intake should also be monitored. Lipid formulas containing EFAs can be based on soybean oil, vegetable-based oil, fish oil or their combination. In parenteral nutrition, 10% or, preferably, 20% lipid emulsion, is available on the market. The optimal lipid E delivery percentage should be between 25 and 50% of nonprotein calories [38] (around 20–30% of whole E delivery). In the parenteral form, the recommended dosing is between 1.0 and 3.0 g/kg/day (in preterm and term infants up to 4.0 g/kg/day) of composite 20% lipid emulsion [38]. Nutrition markers, especially liver biomarkers and serum triglycerides levels, should be monitored in the case of lipid administration. In the case of plasmatic triglyceride level elevation over 3 mmol/Lin infants or 4.5 mmol/L in older children, lipid intake should be decreased [38].

## 6. Fluid, Electrolytes

Fluid in the form of infusion therapy is a part of intensive care management with the aim of solving potential dehydration or hypovolemia, or as a part of hemodynamic optimization. Fluid as a part of nutritional support/intervention is prescribed to maintain adequate hydration and replenish the daily fluid losses. The visible and invisible daily fluid losses might significantly differ in between patients based on many contributing factors. Daily fluid and electrolyte intake should therefore be individualized based on the patient’s actual clinical condition [39]. The Holiday and Segar formula for the calculation of maintenance water needs based on the patient´s weight can be used for initial basic guidance [40]—see Table 3. Isotonic-balanced crystalloid solutions should be the preferred choice. Another possible maintenance fluid estimation based on age is as follows: <1 year (120–150 mL/kg/day), 1–5 years (80–120 mL/kg/day), 6–18 years (50–80 mL/kg/day) [39]. If a balanced crystalloid solution cannot be used (e.g., fluid restriction), basal daily intake of electrolytes should be supplemented with Na and K (1–3 mmol/l/day) as well as Cl (2–4 mmol/l/day) [39]. When prescribing fluids for a critically ill child, care should be taken to respect the present volume and hydration status of the patient (especially hypervolemia) and take into consideration the amount of so-called “junk fluids”, e.g., fluids used as medication (especially antibiotics and other) vehicles and other intravenous fluids. Hyperhydration should be always avoided.

## 7. Micronutrients

Trace elements and vitamins are essential for many biochemical processes and metabolic cycles. In patients on an oral diet or enteral nutrition, complex enteral feeding formulas, or even mixed hospital nutrition (ideally prescribed in cooperation with the nutrition support professionals), contain adequate amounts of trace elements (e.g., Cu, Fe, Zn, Se and others) and vitamins (water- and fat-soluble). Although vitamin C supplementation in critical illness in particular has been recently broadly investigated with slightly promising results [41,42], routine supplementation is currently not recommended outside of clinical trials. On the other hand, in patients on parenteral nutrition, the provision of essential trace elements and vitamins is vital for proper metabolism function. Complex multivitamins are available on the market, however, with different regional availability. The dosing and administration should be based on the manufacturer’s recommendation, preferably by a prolonged infusion (up to 24 h). Recommended dosing of vitamins and trace elements by ESPEN (level of expert opinion) is listed in Table 4 [43,44].

## 8. Enteral Nutrition

Enteral nutrition is a preferred option of nutrition support in critically ill pediatric patients unless contraindicated. Enteral feeding (also in small dosing) has a positive effect on gastrointestinal mucosal integrity and a trophic effect on enterocytes [1]. The most natural route of intake is oral, if possible. Patients in the PICU should therefore be evaluated on daily basis, and oral intake should be encouraged as well as early support of breastfeeding. In the case of consciousness disorder, deep sedation, or the inability to swallow or drink, a gastric tube (preferentially nasogastric) can be used for specific enteral nutrition formula or (fortified) breast milk delivery. In the case of gastric feeding intolerance or a severe risk of aspiration, a possible option is post-pyloric feeding (post-pyloric feeding tube inserted and floated further in the jejunum or inserted by an endoscopist). In chronic intensive care (a stay of 6 weeks or longer), a possible route for enteral feeding could be the percutaneous gastrostomy (PEG) or jejunostomy (PEG-J or surgical jejunostomy).

There are several methods for enteral nutrition administration: continuous feeding involves administration of EN over 24 h assisted by a feeding pump; cyclic feeding involves administration of EN over a time period of <24 h generally assisted by a feeding pump; intermittent feeding involves administration of EN over 20–60 min every 4–6 h via pump assist or gravity assist; and bolus feeding involves administration of EN over a 4 to 10 min period using a syringe (5–6 times per day, up to 7–8 times in small infants) [45]. The continuous or cyclic application is preferred in the case of insufficient enteral feeding tolerance. The aim is to reach the estimated E and protein delivery in a stepwise approach [1] 2–3 days after PICU admission [4], as ideally defined in implemented local-based nutrition/feeding protocol [1]. For a possible stepwise approach to enteral nutrition, see Figure 1. 

The aim of enteral feeding is to reach at least 2/3 of target E delivery during the first 7 days [1]. The standard polymeric enteral formula can be used initially in the majority of patients. In patients not tolerating enteral feeding, an oligomeric or peptide-based formula can improve the tolerance [1]. The majority of nutrition formulas on the market are isocaloric (1 kcal/mL). The dense/hypercaloric formulas (>1 kcal/mL) can be used to limit to total daily enteral feeding volume. Breast milk should be preferentially used to cover the energy and nutrient needs of very small critically ill infants. In the case of absence of breast milk, specific hypocaloric (around 0.65–0.7 kcal/mL) infant formulas or formula milk is to be used. Routine gastric residual volume (GRV) assessment in enterally fed critically ill children is not recommended for enteral feeding tolerance evaluation anymore [1,4,18], but nevertheless can still be implemented in the regional feeding protocols. Enteral feeding intolerance is a clinical condition diagnosed by the intensive care nurse and/or clinician based on the presence of nausea/vomiting, abdominal distension, abdominal discomfort, diarrhea or GRV > 3 mL/kg (if used, in ≥2 subsequent measurements). When dealing with enteral feeding intolerance, several methods are available, and clinicians should follow their local nutrition protocol. Available methods to improve potential enteral intolerance are listed in Table 5. In all patients on full enteral feeding, a minimal protein daily intake of ≥1.5 g/kg/d should be reached. If this is not achieved with the standard enteral formula, additional protein powders should be added.

## 9. Parenteral Nutrition

Parenteral nutrition (PN) is associated with higher morbidity, infectious complications and mortality in comparison with enteral nutrition [11,12]. PN can be individually initiated during the first week of PICU stay in children unable to receive any enteral nutrition; however, supplemental PN should be delayed after the first 7 days (in patients on enteral nutrition) [1,18]. When considering supplemental PN, the content must be individualized daily according to protein and E targets and the amount of enterally delivered feeding tolerance. On the other hand, total parenteral nutrition should involve complete E delivery and protein delivery based on ideally mixed protein, carbohydrates, lipids ratio (e.g., 25:50:25% of energy content) together with trace minerals, vitamins and electrolytes. So-called peripheral parenteral nutrition can be administered through the peripheral venous cannula, due to osmolarity <850 mosmol (does not irritate the vein walls, lower risk of phlebitis). Low osmolarity is derived from a glucose/dextrose concentration below 10%, a lower concentration of nutrients and sometimes a lack of lipid emulsion in the mix (therefore, these types of PN cannot be used for total parenteral nutrition). Total PN, due to high osmolarity, should be administered through a central vein catheter (CVC), peripherally inserted central venous catheter (PICC), tunneled central venous catheter or implanted port system in the central vein [46]. When prescribing total PN, individually based nutrition (all parts of the system—macronutrients and micronutrients being prescribed by the clinician, prepared mainly by the hospital pharmacy), or commercially based/standard (prepared by the pharmaceutic companies) can be chosen. According the recent recommendations, standard (commercially available default all-in-one preparations) should be preferred in the majority of patients (including newborns), if possible, to reach the individual nutritional requirements, with individually based PN being another option [47]. If standard parenteral nutrition is used, micronutrients should also be added using the prepared mixtures of water- and lipid-soluble vitamins and trace elements. The specifics of prematurely born infants need to be taken into account, and additional solutions might be needed to cover their specific needs, e.g., higher calcium and phosphorus demands.

## 10. Examples of Enteral and Parenteral Feeding Prescriptions in PICU

Example No.1: 6-year-old male patient without previous comorbidities, admitted to PICU for septic shock, intubated and mechanically ventilated.

Nutrition screening and anthropometric measurement—(e.g., Strongkids):
Low risk of malnutrition, weight = 25 kg, height 120 cm.Indirect calorimetry or Schofield equation E and protein target calculation:
E target: 1061 kcal/d, protein target at least: 37.5 g/d.Gastric tube insertion, and after initial stabilization (fluid resuscitation, antibiotics), gastric nutrition started according to local protocol within 24 h from admission:
Isocaloric enteral nutrition formula (1 kcal/1 mL, target 1061 mL/d), bolus form 5 × times per day—initial bolus 10% of estimated dose = 1061/5 times per day/10% ≈ 20 mL of initial dose, 40 mL of second dose, 60 mL of third dose, in the case of intolerance (GRV > 3 mL/kg, dose reduction and optimize to tolerance).

Example No. 2: 11-year-old female patient with severe decompensated Crohn’s disease, after being admitted to PICU after subtotal colectomy, low risk of refeeding syndrome.

Nutrition screening and anthropometric measurement—(e.g., Strongkids):
High risk of malnutrition, weight = 35 kg, height 140 cm.Indirect calorimetry or Schofield equation E and protein target calculation:
E target: 1276 kcal/d, protein target at least: 52.5 g/d.Gastric tube in situ, severe abdominal distension, enteral nutrition according to the surgeon currently contraindicated:
Total parenteral nutrition (with added micronutrients) via central/peripherally inserted central line with frequently reassessment of the possibility of “trophic” enteral feeding.If individual PN—53 g of 10% amino acids (525 mL) + 36 g of 20% lipid emulsion (180 mL) + 178 g of 40% glucose (445 mL) + 40 mL of 10% NaCl + 35 mL of 7.5% KCl + 5 mL MgSO4 10% + 5 mL of Ca gluconicum + multivitamin + trace elements + 640 mL of aqua pro injectione (Holiday and Segar formula fluids 1875 mL/day).

## 11. Nutritional Status Evaluation during PICU Stay

During the PICU stay, nutritional status (anthropometric, weight gain/loss) and laboratory parameters (electrolytes, albumin, prealbumin, serum protein levels, serum transferrin, triglycerides and other lipids) should be closely monitored. In patients on enteral feeding, the tolerance should be frequently evaluated, with the aim of switching the gastric enteral feeding to a peroral diet as soon as possible. In patients on PN, daily re-evaluation of the enteral feeding should be done, because even a small (so-called trophic) dose of enteral nutrition can have a positive impact on patients’ overall morbidity and mortality (gut mucosa integrity, nutritional effect on enterocytes).

## 12. Conclusions

Nutritional support remains one of the keystones of intensive care in adult and pediatric patients. Nutrition status evaluation and early nutritional support (preferentially by enteral route) with a protocol-based approach is considered the standard of care. Enteral nutrition is preferred to parenteral whenever possible. Oral intake should be re-initiated as soon as feasible. In infants, the breastfeeding and/or expressed breast milk intake should be supported; in fact, it may be used as the exclusive source of nutrition.

## Figures and Tables

**Figure 1 children-09-01031-f001:**
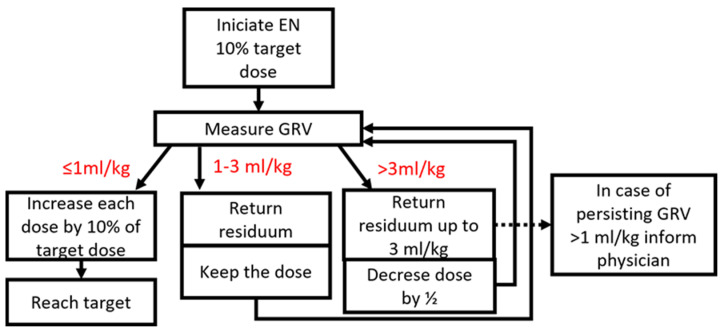
Example of a stepwise approach to enteral nutrition.

**Table 1 children-09-01031-t001:** Energy delivery calculation equations with model examples.

Schofield Equation (kcal/Day)			
males	<3 years	REE = 0.167 × W + 15.174 × H − 617.6	598.59
	3–10 years	REE = 19.59 × W + 1.303 × H + 414.9	950.03
	10–18 years	REE = 16.25 × W + 1.372 × H + 705.8	1865.29
females	<3 years	REE = 16.252 × W + 10.232 × H − 413.5	567.58
	3–10 years	REE = 16.969 × W + 1.618 × H + 371.2	888.56
	10–18 years	REE = 8.365 × W + 4.65 × H + 200	1467.31
**Food Agriculture Organization (MJ/D)**			
males	<3 years	REE = 0.255 × W − 0.226	545.89
	3–10 years	REE = 0.0949 × W + 2.07	948.37
	10–18 years	REE = 0.0732 × W + 2.72	1647.32
females	<3 years	REE = 0.255 × W − 0.214	558.32
	3–10 years	REE = 0.0941 × W + 2.09	949.33
	10–18 years	REE = 0.051 × W + 3.12	1440.49
**World Health Organization (WHO)**			
males	<3 years	60.9 × W − 54	555
	3–10 years	22.7 × W + 495	949
	10–18 years	17.5 × W + 651	1648.5
females	<3 years	61 × W − 51	559
	3–10 years	22.5 × W + 499	949
	10–18 years	22.2 × W + 746	2011.4

REE—resting energy expenditure, kcal—kilocalorie, MJ—megajoule, 1 kcal = 4184 × 103 MJ, W—weight in kilograms, H—height in centimeters. Examples of REE in kcal of three model patients can be seen in the third column: 10 kg 80 cm, 20 kg 110 cm, 57 kg 170 cm, respectively.

**Table 2 children-09-01031-t002:** Amino acid classification by the possibility of synthesis in vivo.

Essential	Semi-Essential (Conditionally Essential)	Nonessential
Arginine	Cysteine	Alanine
Histidine	Glutamine	Asparagine
Isoleucine	Hydroxyproline	Aspartate
Leucine	Proline	Glutamate
Lysine	Taurine	Glycine
Methionine		Serine
Phenylalanine		Tyrosine
Threonine		
Tryptophan		
Valine		

**Table 3 children-09-01031-t003:** Holiday and Segar formula for maintenance fluid calculation.

Weight	mL/kg/d	mL/kg/h
A: the first 10 kg	100	4
B: between 10 and 20 kg	+50 mL/kg/d	+2 mL/kg/h
C: any kg above 20 kg	+25 mL/kg/d	+1 mL/kg/h
Daily calculation	A + B + C	A + B + C

Example: 22 kg patient = (first 10 kg × 100 mL/kg/d) + (second 10 kg × 50 mL/kg/d) + (remaining 2 kg × 25 mL/kg) = 1000 mL + 500 mL + 50 mL = 1550 mL/day. Should be used only for guidance and individualized by patients’ clinical condition and diagnosis! Abbreviations: mL/kg/d—milliliters per kilogram of patient weight per day, mL/kg/h—milliliters per kilogram of patient weight per hour.

**Table 4 children-09-01031-t004:** Recommended dosing of vitamins and trace elements by the European Society of Clinical Nutrition and Metabolism (ESPEN).

Fat-Soluble Vitamins (Vitamin A, D, E, K)	
Vitamin A	150–300 µg/kg/d
Vitamin D	40–150 IU/kg/d up to 400–600 IU/d
Vitamin E	2.8–3.5 mg/kg/d or 2.8–3.5 IU/kg/d 11 mg/d or 11 IU/d
Vitamin K	10 µg/kg/d (or 200 µg/d
**Water-soluble vitamins (Vitamin C, B vitamins)**	
Vitamin C	15–25 mg/kg/d up to 80 mg/d
Vitamin B1 (Thiamine)	0.35–0.50 mg/kg/d up to 1.2 mg/d
Vitamin B2 (Riboflavin)	0.15–0.2 mg/kg/d up to 1.4 mg/d
Vitamin B3 (Niacin)	4–6.8 mg/kg/d up to 17 mg/d
Vitamin B5 (Pantothenic acid)	2.5 mg/kg/d up to 5 mg/d
Vitamin B6 (Pyridoxine)	0.15–0.2 mg/kg/d up to 1.0 mg/kg/d
Vitamin B7 (Biotin)	5–8 µg/kg/d up to 20 µg/d
Vitamin B9 (Folic acid)	56 mg/kg/d up to 140 mg/d
Vitamin B12 (Cyanocobalamin)	0.3 µg/kg/d up to 1 µg/d
**Trace minerals/elements**	
Iron	50–250 µg/kg/d up to 5 mg/d
Zinc	50–500 µg/kg/d up to 5 mg/d
Copper	20–40 µg/kg/d up to 0.5 mg/d
Iodine	1–10 µg/kg/d
Selenium	2–7 µg/kg/d up to 100 mg/d
Manganese	≤1 µg/kg/d up to 50 mg/d
Molybdenum	0.25–1 µg/kg/d up to 5 mg/d

Adopted from [43,44].

**Table 5 children-09-01031-t005:** Methods to improve enteral feeding intolerance.

1. Prokinetics	Metoclopramide, Domperidone—agents for gastric motility improvement
2. Continuous enteral feeding via enteral feeding pump	Usually, 18–19 h of continuous administration via gastric tube with 5–6 h pause
3. Bowel stimulation	Erythromycin—stimulate the bowel motility Consider subcutaneous naltrexone in patients on opioidsSuppository rectally applied
4. Oligomeric formula	Oligomeric formula or peptide-based formula
5. Post-pyloric feeding	Jejunal tube placement and continuous enteral feeding without the night pause

## Data Availability

This review does not report any study data.

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
