# Peer review of "Nutrition in Pediatric Intensive Care: A Narrative Review"

_children, 2022, doi:10.3390/children9071031_

Round 1
Reviewer 1 Report
Synopsis:
This is a review of literature related to nutrition in pediatric intensive care. The aim of this narrative review is to describe the stet-by-step approach for nutritional support of the general population of critically ill pediatric patients in the PICU.
Concern:
This is a well organized and thoughtful review of the current literature. My only recommendation would be to expand the section entitled, Nutritional status assessment, which mentions only one pediatric nutrition screening tool (STRONGkids). It would be appropriate to strengthen this initial step in the nutritional support of critically ill pediatric patients in the PICU by discussing other nutrition screening tools and distinguishing these tools from a formal detailed nutritional assessment. Nutrition screening tools are expected to be able to be administered quickly (i.e., less than 15 minutes) and do not require special training for the caregiver administering the screening tool. Nutritional assessment generally requires special training, clinical expertise, and more time to complete. Such a formal nutritional assessment typically requires a dietitian or other qualified health care provider. The guidelines from ASPEN and SCCM recommend that a detailed nutritional assessment be completed for every patient admitted to the PICU within 48 hours of admission. Since it is not always possible for every patient to undergo such a formal nutritional assessment in a timely manner, the guidelines further state that “a validated method to screen critically ill children for malnutrition risk may help allocate resources to high-risk patients.”
I have listed several other pediatric nutrition screening tools (1, 2, 3, 4) although the list is not exhaustive. To my knowledge, no pediatric nutrition screening tool has been validated in the PICU patient population. Additionally, a recent multicenter study of inpatients in 14 hospitals across 12 European countries found that three of these more commonly used screening tools showed a fair amount of variability in classification of malnutrition risk and each tool failed to identify a proportion of children with subnormal anthropometric measures (5). This indicates that screening is not an adequate substitute for a formal, detailed nutritional assessment.
1. Sermet-Gaudelus I, Poisson-Salomon AS, Colomb V, Brusset MC, Mosser F, Berrier F, Ricour C. Simple pediatric nutritional risk score to identify children at risk of malnutrition. Am J Clin Nutr 2000;72:64-70.
2. Gerasimidis K, Macleod I, Maclean A, Buchanan E, McGrogan P, Swinbank I, McAuley M, Wright CM, Flynn DM. Performance of the novel paediatric Yorkhill malnutrition score (PYMS) in hospital practice. Clinical Nutrition 2011;30:430-435.
3. McCarthy H, Dixon M, Crabtree I, Eaton-Evans MJ, McNulty H. The development and evaluation of the Screening Tool for the Assessment of Malnutrition in Paediatrics (STAMP©) for use by healthcare staff. J Hum Nutr Diet. 2012;25:311-318.
4. White M, Lawson K, Ramsey R, Dennis N, Hutchison Z, Soh XY, Matsuyama M, Doolan A, Todd A, Elliott A, Bell K, Littlewood R. Simple nutrition screening tool for pediatric inpatients. JPEN J Parenter Enteral Nutr. March 2016;392-398. DOI: 10.1177/0148607114544321
5. Chourdakis M, Hecht C, Gerasimidis K, Joosten KFM, Karagiozoglou-Lampoudi T, Koetse HA, Ksiazyk J, Lazea C, Shamir R, Szajewska H, Koletzko B, Hulst JM. Malnutrition risk in hospitalized children: use of 3 screening tools in a large European population. Am J Clin Nutr. 2016;103:1301-1310.
Author Response
We are grateful for this valuable comment and we agree that the initial screening of the nutritional status is of the utmost importance. We added other examples of the nutritional screening tools and expanded the related literature. We also added an explanation concerning the differences between screening and (further) complete nutritional status assessment. We also agree that the complete assessment within 48 hours in every PICU patient is probably not accomplishable but that it is important in certain subgroups of (previously identified) high-risk patients.
Reviewer 2 Report
Milan et al present a narrative review concerning nutritional support in pediatric intensive care. Such papers are welcome to highlight the importance of nutrition in this population and the need for further studies on this field. Overall, the review is interesting and comprehensive. I have only minor comments:
In line 84: “anthropometrics” instead of “antropomethrics”
Lines 127-129: According to the referenced ESPEN guidelines this is recommeded only if IC is used (see recommendations 16-18) . If predictive equations are used to estimate the energy need, hypocaloric nutrition (below 70% estimated needs) should be preferred over isocaloric nutrition for the first week of ICU stay (Recommendation 19).
Line 175: “is limited by” instead “is based on”
Lines 211-212: “In parenteral form the recommended dosing is…” instead of “In parenteral form is the recommended dosing…”
Table 3: abbreviations should be provided
Lines 278-279: Please rephrase or explain further providing the appropriate reference(s): Continuous feeding involves hourly administration of EN over 24 hours assisted by a feeding pump. On the other hand cyclic feeding involves administration of EN over a time period of <24 hours generally assisted by a feeding pump (Nutr Clin Pract. 2018 Dec;33(6):790-795. doi: 10.1002/ncp.10105.).
Line 317: “..osmolarity <850 mOsmol/L…” instead of “…osmolality <850mosmol…”
Line 318: ”…The low osmolarity derives from…” instead of “…The low osmolality is caused by…”
Line 321: “osmolarity” instead of “osmolality”
Author Response
We thank the reviewer for the valuable comments.
We adjusted the wording of several sentences according to the reviewers suggestions.
We added abbreviations explanations to the Table 3.
We slightly rephrased the wording concerning the ESPEN recommendations for the isocaloric vs. hypocaloric feeding.
We rephrased the sentence concerning the cyclic feeding using enteral pump according to the reference provided by the reviewer.